

**Dynamics of transparent exopolymer particles (TEP) during**
**the VAHINE mesocosm experiment in the New Caledonia**
**lagoon**
**I. Berman-Frank[1], D. Spungin[1], E. Rahav[1,2], F. Van Wambeke[3], K. Turk-Kubo[4], T.**
**Moutin[3]**
[1] Mina and Everard Goodman Faculty of Life Sciences, Bar-Ilan University, Ramat Gan,
Israel 5290002
[2] National Institute of Oceanography, Israel Oceanographic and Limnological Research,
Haifa, Israel
[3] Aix Marseille Université, CNRS/INSU, Université de Toulon, IRD, Mediterranean
Institute of Oceanography (MIO) UM110, 13288, Marseille, France
[4] Ocean Sciences Department, University of California, Santa Cruz, 1156 High Street, Santa
Cruz, CA, 95064, USA
Correspondence to: I. Berman-Frank (ilana.berman-frank@biu.ac.il)



**Abstract**
In the marine environment, transparent exopolymeric particles (TEP) produced from abiotic
and biotic sources link the particulate and dissolved carbon pools and are essential vectors
enhancing vertical carbon flux. We characterized spatial and temporal dynamics of TEP
during the VAHINE experiment that investigated the fate of diazotroph derived nitrogen and
carbon in three, replicate, dissolved inorganic phosphorus (DIP)-fertilized 50 m$^3$ enclosures in
an oligotrophic New Caledonian lagoon. During the 23 days of the experiment, we did not
observe any depth dependent changes in TEP concentrations in the three sampled-depths (1,
6, 12 m). TEP carbon (TEP-C) content per mesocosm averaged $28.9 \pm 9.3\%$ and $27.0 \pm 7.2\%$
of TOC in the mesocosms and surrounding lagoon respectively and was strongly and
positively coupled with TOC during P2. TEP concentrations declined for the first 9 days after
DIP fertilization (P1 = days 5-14) and then gradually increased during the second phase (P2 =
days 15-23). Temporal changes in TEP concentrations paralleled the growth and mortality
rates of the diatom-diazotroph association of *Rhizosolenia* and *Richelia* that predominated the
diazotroph community during P1. By P2, increasing total primary and heterotrophic bacterial
production consumed the supplemented P and reduced availability of DIP. For this period,
TEP concentrations were negatively correlated with DIP availability and turnovertime of DIP
($T_{DIP}$) while positively associated with enhanced alkaline phosphatase activity (APA) that
occurs when the microbial populations are P-stressed. During P2, increasing bacterial
production (BP) was positively correlated with higher TEP concentrations which were also
coupled with the increased growth rates and aggregation of the unicellular UCYN-C
diazotrophs which bloomed during this period. We conclude that the composite processes
responsible for the formation and breakdown of TEP yielded a relatively stable TEP pool
available as both a carbon source and facilitating aggregation and flux throughout the
experiment.  TEP was probably mostly influenced by abiotic physical processes during P1
while biological activity (BP, diazotrophic growth and aggregation, export production) mainly
impacted TEP concentrations during P2 when DIP-availability was limited.

**1  Introduction**
The cycling of carbon (C) in the oceans is a complex interplay between physical,
chemical, and biological processes that regulate the input and the fate of carbon within the
ocean. An essential process driving the flux of carbon and other organic matter to depth and
enabling long term sequestration and removal of carbon from the atmosphere is the biological



pump that drives organic C formed during photosynthesis to the deep ocean. This process, termed export production (Eppley and Peterson, 1979), is facilitated via physical inputs of 'new' nutrients (e.g. nitrogen, phosphorus, silica, trace metals, etc.) into the euphotic zone from either external sources (deep mixing of upwelled water, river discharge, dust deposition, and anthropogenic inputs) or via biological processes such as microbial $N_2$ fixation that converts biologically unavailable dinitrogen ($N_2$) gas into bioavailable forms of nitrogen and enhances the productivity of oligotrophic oceanic surface waters that are often limited by nitrogen (Falkowski, 1997; Capone, 2001).

Marine $N_2$ fixation is performed by diverse prokaryotic organisms comprised predominantly of autotrophic cyanobacteria and heterotrophic bacteria (Zehr and Kudela, 2011). To supply the energetically-expensive process of converting $N_2$ to ammonia (Stam et al., 1987; Postgate and Eady, 1988; Mulholland and Capone, 2000), these organisms must obtain energy from either photosynthesis (cyanobacteria) or from bioavailable organic carbon compounds within the aquatic milieu (heterotrophic bacteria and mixotrophs). The total organic carbon (TOC) in the ocean contains dynamic particulate (POC) and dissolved organic carbon (DOC) pools that are supplied by biotic sources that are broken down into organic C-containing marine microgels which include transparent polymeric particles (TEP). TEP are predominantly acidic polysacchridic organic particles ranging in size from ~0.45 to > 300 μm and are found in both marine and freshwater habitats (Passow, 2002). Both biotic and abiotic processes form aquatic TEP that are routinely detected by staining with Alcian Blue (Alldredge et al., 1993; Passow and Alldredge, 1995). Abiotic TEP occur by coagulation of colloidal precursors in the pool of dissolved organic matter (DOM) and from planktonic debris (Passow, 2002; Verdugo and Santschi, 2010) that may be stimulated by turbulence or by bubble adsorption (Logan et al., 1995; Zhou et al., 1998; Passow, 2002). Biotically TEP form from extracellular excretion or mucilage in algae and bacteria and from grazing and microbial breakdown of larger marine snow particles [reviewd in (Passow, 2002; Bar-Zeev et al., 2015)].

TEPs are light and bouyant (Azetsu-Scott and Passow, 2004). Yet, once formed, TEPs sticky nature enhances and consolidates the formation of larger aggregates such as marine/lake snow providing favorable environments for diverse microorganisms (Passow, 2002; Engel, 2004). Sedimentation of TEP associated "hot spots" from the surface are important for transporting particulate organic material and microorganisms to deeper waters (Smith and Azam, 1992; Azam and Malfatti, 2007; Bar-Zeev et al., 2009). During



sedimentation, TEP can also function as a direct source of carbon and other nutrients for
higher trophic level organisms such as protists, micro-zooplankton, and nekton (Passow,
2002; Engel, 2004).

TEP production can be enhanced in late phases of algal blooms and in scenescent or

nutrient-stressed phytoplankton (Grossart et al., 1997; Passow, 2002; Engel, 2004;
Berman-Frank et al., 2007). Thus, TEP in oligotrophic waters (Engel, 2004) provide a source
of available carbon to fuel microbial food webs (Azam and Malfatti, 2007) that typically
succeed autotrophic blooms. TEP based aggregates or marine snow containing TEP typically
with high carbon (C): nitrogen (N) ratios (Wood and Van Valen, 1990; Berman-Frank and
Dubinsky, 1999), which can also fuel $N_2$ fixation by heterotrophic diazotrophs both in
oxygenated surface waters and in the aphotic zones (Rahav et al., 2013; Benavides et al., in
press).
The VAHINE project was designed to examine the fate/s of 'newly'-fixed N by
diazotrophs or diazotroph-derived N (hereafter called DDN) in the pelagic food web using
large mesocosms in the oligotrophic tropical lagoon of New Caledonia where diverse
diazotrophic populations have been observed (Dupouy et al., 2000; Garcia et al., 2007; Rodier
and Le Borgne, 2008; Biegala and Raimbault, 2008; Rodier and Le Borgne, 2010; Bonnet et
al., This issue-b). One of the major questions addressed during VAHINE was whether
diazotroph blooms significantly modify the stocks, fluxes, and ratios of biogenic elements (C,
N, P, Si) and the efficiency of carbon export. To this end, the 3 large-volume (~50 m$^3$)
mesocosms containing ambient lagoon waters were fertilized with 0.8 µM DIP, and multiple
parameters were measured inside and outside of the mesocosms for 23 days (details of
parameters and experimental setup in (Bonnet et al., This issue-b). Within the VAHINE
framework, our specific objectives were: 1) to examine the spatial and temporal dynamics of
TEP; 2) to determine whether TEP content was regulated by nutrient status in the mesocosms
- specifically DIP availability; 3) to examine the relationship between TEP content, particulate
and dissolved carbon, and primary or heterotrophic bacterial production; and 4) to elucidate
whether TEP provided a source of energy for diazotrophs/bacteria/mixotrophs in mesocosms.

**2   Methods**
**2.1   Study site, mesocosm description, and sampling strategy**
Three large-volume (~50 m$^3$) mesocosms were deployed at the exit of the oligotrophic
New Caledonian lagoon (22º29.10 S–166º26.90 E), from 13 January 2013 (day 1) to 4



February 2013 (day 23). The complete description of the mesocosm design and deployment,
as well as sampling strategy is detailed in Bonnet et al. (This issue-b). The mesocosms were
intentionally supplemented with 0.8 µmol $KH_2PO_4$ (hereafter referred to as DIP fertilization)
between day 4 and 5 day of the experiment to promote $N_2$ fixation. Samples were collected
during the early morning of each day for 23 days with a clean Teflon pumping system from 3
selected depths (1 m, 6 m, 12 m) in each mesocosm (M1, M2 and M3) and outside (hereafter
called 'lagoon waters'-O). Based on the results of different biogeochemical and biological
parameters during VAHINE (Turk-Kubo et al., 2015; Berthelot et al., 2015; Bonnet et al.,
This issue-a), three specific periods were discerned within which we have also investigated
TEP dynamics: Days 2-4 (P0) are the pre-fertilization days; days 5-14 (P1), and days 15-23
(P2).

## 2.2  TEP quantification

Water samples (100 mL) were gently (< 150 mbar) filtered through a 0.45 µm
polycarbonate filters (GE Water & Process Technologies). Filters were then stained with a
solution of 0.02% Alcian Blue (AB) and 0.06% acetic acid (pH of 2.5). The excess dye was
removed by a quick deionized water rinse. Filters were then immersed in sulfuric acid (80%)
for 2 h, and the absorbance at 787 nm was measured spectrophotometrically (CARY 100,
Varian). AB was calibrated using a purified polysaccharide GX (Passow and Alldredge,
1995). TEP concentrations (µg gum xanthan [GX] equivalents $L^{-1}$) were measured according
to (Passow and Alldredge, 1995). Total TEP content in the mesocosms was calculated by
integrating the weighted average of the TEP concentrations per depth and multiplying by the
specific volume of each mesocosm. To estimate the role of TEP in C cycling, total amount of
TEP-C was calculated for each mesocosm, using the volumetric TEP concentrations at each
depth, the specific volume per mesocosm, and the conversion of GX equivalents to carbon
applying the revised factor of 0.63 based on empirical experiments from both natural samples
from different oceanic areas and phytoplankton cultures (Engel, 2004).

## 2.3  TOC, POC, DOC

Samples for total organic carbon (TOC) concentrations were collected in duplicate from
6 m in each mesocosm and in lagoon waters in precombusted sealed glassware flasks,
acidified with $H_3PO_4$ and stored in the dark at 4 °C until analysis. Samples were analyzed on a
Shimadzu TOCV analyzer with a typical precision of 2 µmol $L^{-1}$. Samples for particulate
organic carbon (POC) concentrations were collected by filtering 2.3 L of seawater through a





precombusted GF/F filter (450 °C for 4 h), combusted and analyzed on an EA 2400 CHN
analyzer. Dissolved organic carbon (DOC) concentrations were calculated as the difference
between TOC and POC concentrations. Fully detailed methodologies and data are available in
Berthelot et al. (2015).

## 2.4   Dissolved inorganic phosphorus (DIP) and alkaline phosphatase activity (APA)

The determination of DIP concentrations are detailed in Berthelot et al. (2015). Samples
for DIP were collected from each of the three depths in M1, M2 and M3 and lagoon waters
(O) in 40 mL glass bottles, and stored in -20 °C until analysis. DIP concentration was
determined using a segmented flow analyzer according to (Aminot and Kérouel, 2007). The
alkaline phosphatase activity (APA) was measured from the same depths and sites using the
analog substrate methylumbelliferone phosphate (MUF-P, 1 µM final concentration;
SIGMA), (Hoppe, 1983). Full details of the measurements and analyses are described in Van
Wambeke et al. (This issue).

## 2.5   Chlorophyll a (Chl *a*), Primary production (PP) and DIP turnover time

Chlorophyll a (Chl *a*) concentrations were determined by fluorimetry and the detailed
methodologies also for primary production are described in Berthelot et al. (2015). Briefly,
primary production (PP) rates and DIP turnover time ($T_{DIP}$, i.e., the ratio of $PO_4^{-3}$
concentration and uptake) were measured using the $^{14}C/^{33}P$ dual labeling method (Duhamel et
al., 2006). 60 mL bottles were amended with $^{14}C$ and $^{33}P$ and incubated for 3 to 4 h. This was
followed by the addition of 50 µL of $KH_2PO_4$ solution (10 mmol $L^{-1}$) to stop $^{33}P$ assimilation.
Samples were kept in the dark to stop $^{14}C$ uptake. Samples were filtered on 0.2 µm
polycarbonate membrane filters, and counts were done using a Packard Tri-Carb® 2100TR
scintillation counter. PP and $T_{DIP}$ were calculated according to (Moutin et al., 2002).

## 2.6   Bacterial production (BP)

Heterotrophic bacterial production (BP) was estimated using the $^3H$-leucine
incorporation technique (Kirchman, 1993), adapted to the centrifuge method (Smith and
Azam, 1992). The complete methodology including enumeration of heterotrophic bacterial
abundances (BA) by flow cytometry is detailed in Van Wambeke et al. (This issue).



### 2.7 N$_2$ fixation, diazotrophic abundance and growth rates


N$_2$ fixation rates were determined daily on ambient waters from mesocosms and the
lagoon. Samples were spiked with 99% $^{15}$N$_2$-enriched seawater, incubated in-situ under
ambient light and seawater temperatures as detailed in Berthelot et al. (2015) and (Bonnet et
al., This issue-a).

Data and protocols of sampling for diazotrophic abundance and calculation of their

respective growth rates are detailed fully in Turk-Kubo et al. (2015). Briefly, samples (from 6
m only) were collected every other day from the mesocosms, and from the lagoon waters.
DNA was extracted and nine diazotrophic phylotypes were identified using quantitative PCR
(qPCR). The targeted diazotrophs were two unicellular diazotrophic symbionts of different
*Braarudosphaera bigelowii* strains, UCYN-A1, UCYN-A2; free-living unicellular diazotroph
cyanobacterial phylotypes UCYN-B (*Crocosphaera* sp.), and UCYN-C (*Cyanothece* sp. and
relatives); *Trichodesmium* spp.; and three diatom-diazotroph associations (DDAs), *Richelia*
associated with *Rhizosolenia* (Het-1), *Richelia* associated with *Hemiaulus* (Het-2), *Calothrix*
associated with *Chaetoceros* (Het-3), and a widespread gamma-proteobacterial phylotype γ-
24774A11. Abundances are reported as *nifH* copies L$^{-1}$ as the number of *nifH* copies per
genome in these diazotrophs are uncertain. Growth and mortality rates were calculated for
individual diazotrophs inside the mesocosms when abundances were higher than the limit of
quantification (LOQ) for two consecutive sampling days as detailed in Turk-Kubo et al.

(2015).

### 2.8 Microscopic Analyses


Detailed method for sampling for microscopic analyses is described in Bonnet et al.

(This issue). Phytoplankton were visualized using a Zeiss Axioplan (Zeiss, Jena, 6 Germany)
epifluorescence microscope fitted with a green (510-560 nm) excitation filter, which targeted
the *Richelia* and the UCYN phycoerythrin-rich cells. The diatom-dazotroph association
*Rhizosolenia-Richelia* were imaged in bright-field.

### 2.9 Statistical analyses


Statistical analyses were carried out with XLSTAT, a Microsoft Office Excel based

software. A Pearson correlation coefficient test was applied to examine the association
between two variables (TEP versus physical, chemical, or physiological variable) after linear
regressions or log-transformation of the data. The non-parametric Kruskal–Wallis one-way



analysis of variance was applied to compare between TEP dynamics from each of the
different phases. A confidence level of 95% (α- 0.05) was used.

**3    Results and Discussion**
**3.1    General context and spatial and temporal dynamics of TEP**

The VAHINE experiment was designed to induce and follow diazotrophic blooms and

their fate within an oligotrophic environment (Bonnet et al., This issue-b). Our specific
objectives of investigating TEP dynamics were thus examined within the general context and
aims of the large experiment. The first stage of the experiment involved the enclosure of the
lagoon waters and 3 days of equilibration of the system (P0 – pre-fertilization days 2-4). At
this initial stage the total Chl *a* concentrations averaged around 0.2 μg L$^{-1}$ in the lagoon water
and in the mesocosms and the phytoplankton consisted of diverse representatives from the
cyanobacteria (*Prochlorococcus*, *Synechococcus*, diatoms such as *Pseudosolenia calcar-avis*,
and dinoflagelates (Leblanc et al., This issue). During P0, the most abundant members of the
diazotrophic community in the lagoon waters were *Richelia-Rhizosolenia* (Het-1), the
unicellular UCYN-A1, UCYN-A2, UCYN-C, and the filamentous *Trichodesmium* (Turk-
Kubo et al., 2015).

Fertilization of the mesocosms with DIP on day 4 stimulated a two-stage response by

the diazotrophic community that was further reflected by many of the measured chemical and
biological parameters (Berthelot et al., 2015; Turk-Kubo et al., 2015; Bonnet et al., This
issue-a; Bonnet et al., This issue-b). After fertilization, from day 5 through day 14 (P1),
excluding a significant increase in N$_2$ fixation rates, the functional community-wide
biological responses (Chl *a*, PP, BP, BA) remained relatively low and similar to the values for
P0 and for P1 in the outside lagoon waters (Berthelot et al., 2015; Leblanc et al., This issue;
Van Wambeke et al., This issue). The autotrophic community during P1 was comprised of
picophytoplankton such as *Prochlorococcus*, and *Synechococcus*, micro and
nanophytoplankton including dinoflagellates, and a diverse diatom community (*Chaetoceros*,
*Leptocylindrus, Cerataulina*, *Guinardia,* and *Hemiaulus*), (Leblanc et al., This issue). Diatom-
diazotroph associations (DDAs), predominantly *Richelia-Rhizosolenia* (Het-1) dominated the
diazotroph community in the mesocosms (Turk-Kubo et al., 2015) although it still only
contributed from 2% to ~8% of the total diatom biomass in P0 and P1 respectively (Leblanc
et al., This issue). These DDAs were succeeded during the last 9 days (day 15 to 23 termed



P2) by a large bloom of unicellular diazotrophs characterized predominantly as UCYN-C
(Turk-Kubo et al., 2015).
The final stage of the experiment (P2, days 15-23) was characterized by significantly
enhanced values for many biological parameters including $N_2$ fixation rates, Chl *a*, PP, BA,
BP, and particulate organic carbon and nitrogen compared to their respective average values
in P1 (Leblanc et al., This issue; Van Wambeke et al., This issue; Bonnet et al., This issue-a).
In all three mesocosms, a significant bloom of UCYN-C developed (day 11 – M1, day 13-M2,
day 15-M3) and remained dominant representatives of the diazotroph community until day
23(Turk-Kubo 2015). The ambient autotrophic community responded to the input of new N,
and the transfer of diazotroph derived N was demonstrated and seen in increasing abundance
of *Synechococcus* , pico-eukaryotes, and the non-diazotrophic diatoms *Navicula* and
*Chaetoceros* spp. (Leblanc et al., This issue; Van Wambeke et al., This issue; Bonnet et al.,
This issue-a). Thus the extremely high $N_2$ fixation rates during this experiment provided
sufficient new N to yield high Chl a concentrations (> 1.4 µg L$^{-1}$ ) and rates of PP (>2 µmol C
L-1 d-1)(Berthelot et al., 2015).
**3.1.1 Dynamics of TEP**
TEP concentrations for the entire experimental period ranged from ~22 to 1200 µg GX
L$^{-1}$. In each mesocosm and also in the lagoon waters (O), the TEP concentrations were similar
for the three sampled depths within the 15 m water-column with an overall average of 350 ±
180 µg GX L$^{-1}$ (Fig. S1). Temporally, TEP concentrations generally followed the three
distinct periods (P0, P1, P2) that coincided with the described experimental phases
characterized from the diazotrophic populations and the biogeochemical and biological
(production) parameters (Berthelot et al., 2015; Turk-Kubo et al., 2015; Leblanc et al., This
issue; Van Wambeke et al., This issue; Bonnet et al., This issue-a), (Fig. 1, Fig. S1).
Following the enclosure of the lagoon water in the mesocosms (day 2), TEP concentrations
increased from the lowest volumetric concentrations (averaging~ 50 µg GX L$^{-1}$) measured on
day 2 to reach maximum concentrations in each of the mesocosms (average of ~800 µg GX L$^{-1}$
) on day 5, ~15 h after the mesocosms were fertilized with DIP (Fig. S1, Fig. 1a). From day 5
to day 14 (P1) average TEP content in M2 and M3 decreased slightly yet significantly (p <
0.05) with the major decline in all mesocosms measured from day 5 to 6 (Fig. 1, Fig. S1,
Table S1). From day 15 to 23 (P2) TEP concentrations in all mesocosms increased gradually





($p < 0.05$) over the subsequent 9 days to reach $381 \pm 39$ µg GX $L^{-1}$ on day 23 (Fig. 1, Table
S1).
TEP concentrations in the lagoon waters were compared with those in the mesocosms.
These showed a similar pattern of increase in TEP during P0 and P3 while the gradual decline
in TEP concentrations during P2 was not statistically significant as observed in the
mesocosms (Fig. 1, Fig. S1). In the lagoon waters average TEP concentrations over the whole
experimental period day 2 to day 23 were $335 \pm 56$ µg GX $L^{-1}$. While temporal variations in
the three mesocosms were mostly statistically significant (Fig. 1, Table S1), the total TEP
content calculated for each mesocosm and for an equivalent volume of lagoon water based on
average mesocosm volume) did not differ significantly when we assessed all data obtained
during P1 and P2 (Fig. 2, $p > 0.05$, Kruskal –Wallis analyses of variance). The lack of
significant differences in total TEP content in the mesocosms throughout the experiment
could reflect the contrasting processes of formation and breakdown that together maintain a
relatively stable pool of available TEP.
Mechanical processes such as wave turbulence and tidal effects can influence TEP
formation and breakdown (and resulting content), (Stoderegger and Herndl, 1999; Passow,
2002). Our results indicate no obvious effects of these parameters on TEP content as these
were similar in the enclosed mesocosms and the outside lagoon (Fig. 1, Fig. 2). Moreover,
despite the initial increase in mesocosm TEP concentrations prior to DIP fertilization, and for
the first 15 h after fertilization, from day 5 to the end of the experiment, TEP concentrations
were similar for both DIP-fertilized mesocosms and the lagoon waters with low DIP
concentrations (Fig. 1, Fig. S1, Fig. 2). This implies that also DIP fertilization had no impact
on the resulting total TEP content in the mesocosms (Yet, see below section 3.2).
The relative uniformity and stability of TEP within the 15 m water column of both the
mesocosms and the lagoon waters reflects the homogeneity of the shallow lagoon system. The
variability between the three depths was statistically insignificant in many of the other
physical, chemical, and biological features of the mesocosms and the lagoon waters for
temperature, salinity, inorganic nutrients (N, P, Si), POC, PON, POP, DOC, Chl *a*, and
primary production and heterotrophic bacterial production (Berthelot et al., 2015; Van
Wambeke et al., This issue; Bonnet et al., This issue-b; Bonnet et al., This issue-a). In contrast
to some marine systems where TEP concentrations were correlated with the vertical
distribution of Chl *a* or POC (Passow, 2002; Engel, 2004; Ortega-Retuerta et al., 2009; Bar-
Zeev et al., 2009; Bar-Zeev et al., 2011), the results we obtained here showed no correlation



to the vertical (i.e. depth related) autotrophic signatures. Moreover, the similar TEP
concentrations at 1, 6, and 15 m do not support a sub-surface maxima in TEP concentrations,
stimulated by abiotic aggregation, at the sea-surface top layer as has been reported at 1 m
depth in different oceanic areas (Wurl et al., 2011). Abiotic processes of formation and
breakdown can be influential yet here we do not see a depth-correlated specific abiotic driver
and TEP were evenly distributed within the 15 m water column for all mesocosms (Fig. S1).
**3.2  DIP availability, APA, and TEP content.**
The average TEP concentrations we measured in the New Caledonian waters are
comparable to TEP concentrations reported from other marine environments such as the
eastern temperate-subarctic North Atlantic (Engel, 2004), the Ross Sea (Hong et al., 1997),
western Mediterranean – Gulf of Cadiz and the Straits of Gibraltar (García et al., 2002; Prieto
et al., 2006), the Gulf of Aqaba (northern Red Sea), (Bar-Zeev et al., 2009), in the northern
Adriatic Sea (Radić et al., 2005), and in the New Caledonia lagoon (Mari et al., 2007;
Rochelle-Newall et al., 2008).
While prediction as to the expected TEP concentrations with trophic or productive
status is difficult (Beauvais et al., 2003), decreasing availability of dissolved nutrients such as
nitrate and phosphate have been correlated with enriched TEP concentrations in both cultured
phytoplankton and natural marine systems (Engel et al., 2002; Brussaard et al., 2005; Urbani
et al., 2005; Bar-Zeev et al., 2011). In P-limited systems, low Chl $a$ concentrations often
reflect the nutrient-stressed phytoplankton. As long as light and $CO_2$ are available, limitation
of essential nutrients results in an uncoupling between carbon fixation and growth during
which the excess photosynthate can be used to produce carbon-rich compounds including
TEP (Berman-Frank and Dubinsky, 1999; Mari et al., 2001; Rochelle-Newall et al., 2008).
Moreover, as DIP-availability declines, cells activate P-acquisition pathways and enzymes
such as APA to access P from other sources. Thus, and based on previous data (Bar-Zeev et
al., 2011), we hypothesized that TEP content would be negatively correlated with autotrophic
biomass (Chl $a$) and PP and positively correlated with APA.
Mesocosm fertilization on the evening of day 4 enriched the system with ten-fold
higher DIP concentrations that were available for microbial utilization throughout the
following 8 – 10 days (Berthelot et al., 2015; Van Wambeke et al., This issue; Leblanc et al.,
This issue; Bonnet et al., This issue-b). Thus, when DIP concentrations were relatively
sufficient during P1, no statistically significant relationship was observed between TEP and




POP, DIP, $T_{DIP}$, Chl *a*, or PP (Table S2). This situation changed with the declining availability
of DIP and the shift in the response of the system during P2 from day 15 to 23. During P2
high TEP concentrations were associated with decreasing DIP for each of the mesocosms with
an overall negative correlation ($R^2$ = 0.23, n = 23, p = 0.02), (Fig. 3a). A similar negative
trend was obtained between TEP and the turnover time of DIP ($T_{DIP}$) which can indicate DIP
limitation ($R^2$=0.28 n= 26, p= 0.006), (Fig. 3b).
In the South West Pacific, the critical DIP turnover time ($T_{DIP}$) required for single
filaments of *Trichodesmium* to grow is 2 d (Moutin et al., 2005). Here $T_{DIP}$ values lower than
1 d, indicative of a strong DIP deficiency, were reached on day 14 in M1, day 19 for M2, and
on day 21 for M3 with the average $T_{DIP}$ values during P2 significantly different in each
mesocosm, $T_{DIP}$ of 0.5, 1.8, 3.9 d for M1, M2, M3, respectively (Berthelot et al., 2015). The
deficiency in DIP was reflected in the subsequent APA which increased rapidly in both M1
and M2 from day 18 (average for M1 and M2 during P2 ~8 ± 6 nmol MUF $l^{-1}$ $h^{-1}$) and after
day 21 in M3 illustrating a biological response of the microbial community to P stress (Van
Wambeke et al., This issue). We did not specifically measure TEP production by autotrophic
or heterotrophic plankton. Yet, the significant (although indirect relationship) negative
correlation of TEP with DIP concentrations and $T_{DIP}$ (Fig. 3a-b) suggests that microbial
responses to decreased DIP availability resulted from either 1) an increase in TEP synthesis
through higher polysaccharide production rather than biomass which requires higher nutrients
(Berman-Frank and Dubinsky 1999, (Wood and Van Valen, 1990), or 2) nutrient limitation
inducing greater breakdown of biomass and POM (maybe via programmed cell death) and
subsequent abiotic formation of TEP. We obtained a significant semi-logarithmic relationship
between TEP and APA ($R^2$ = 0.33 n= 25, p = 0.002), (Fig. 3c) which implies active TEP
formation when DIP concentrations are reduced and APA increases until a saturating point
whereby any further increases in APA do not appear to impact TEP concentrations (Fig. 3c).
This relationship may not always be valid as APA in the lagoon waters was consistently
higher at 1 m than APA measured at 6 and 12 m depths (Van Wambeke et al., This issue), yet
TEP concentrations were uniform at all depths (Fig. S1).

### 3.3 TEP and carbon pools

The size range of TEP spans a range of particles from 0.45 to 300 µm (Alldredge et al.,
1993; Bar-Zeev et al., 2015). TEP precursors (0.05 to 0.45 µm size) are formed and broken
down in the DOC pool and thus essentially "TEP establish a bridge between DOM (including



DOC) and the POM pool ""(Engel, 2004). Our data shows a generally stable contribution of
TEP to the TOC pool. Excluding day 5, where TEP-C comprised $56.5 \pm 8\%$ of TOC, the %
TEP-C was $28.9 \pm 9.3\%$ and $27.0 \pm 7.2\%$ of the TOC in all mesocosms and in the lagoon
waters, respectively (Fig. 4a-b).

TEP concentrations can be directly and positively correlated with POC (Engel, 2004)

and with DOC (Ortega-Retuerta et al., 2009). Yet, TEP concentrations can also be negatively
related to POC indicative of low TEP production when POC concentrations are high (Bar-
Zeev et al., 2011). In the mesocosms, a significant positive correlation between TEP
concentrations and TOC was obtained for all three mesocosms only during P2 ($R^2 = 0.75$,
0.73, 0.58 and $p < 0.05$ for M1, M2, M3 respectively), (Fig. 4c, Table S2). This period
coincided with the largest gain in total autotrophic and heterotrophic biomass and elevated $N_2$
fixation, PP, and BP rates (Berthelot et al., 2015; Van Wambeke et al., This issue; Bonnet et
al., This issue-a).

Although TEP was significantly and positively correlated with TOC in the mesocosms

during P2, this was not the case with either POC or DOC in any mesocosm for either P1 or P2
(Table 1).The absence of any significant correlation between TEP and POC was surprising as
TEP are part of the POC pool comprising $40 - 60\%$ of the particulate combined carbohydrates
in POC (Engel, 2004; Engel et al., 2012). Furthermore, we did not obtain any significant
correlations of TEP and specific components of the dissolved organic matter such as
fluorescent dissolved organic matter (FDOM) or chromophoric dissolved organic matter
(CDOM) that was coupled to the dynamics of $N_2$ fixation in the mesocosms (Tedetti et al.,
This issue). The lack of significant correlation could partially reflect methodological issues. In
this experiment [and operationally according to published protocol (Passow and Alldredge
(1995)] TEP was measured on 0.45 μm filters – so that Alcian Blue stained particles included
particles > 0.45 μm while POC was measured on GF/F (nominal pore size 0.7 μm). DOC is
typically considered for the < 0.45 μm fraction (Thurman, 1985), although here no direct
measurements of DOC were made and DOC was obtained by subtracting POC from TOC.
Thus, DOC actually covered the < 0.7 μm fraction. Our methodology therefore precluded
determination of the smaller TEP precursors that would contribute to the DOC and colloidal
pools (Villacorte et al., 2015). As such we probably overestimated TEP relative to POC and at
the same time underestimated TEP's contribution to the DOC pool (Bar-Zeev et al., 2009).
The lacking correspondence between TEP concentrations and the pools of POC and DOC
may also result from the uncoupling between formation and breakdown processes. Abiotic



processes, will modify relationships obtained between biotic TEP production and recycling
(Wurl et al., 2011). Thus, it is feasible that especially during P1 abiotic factors predominated
breaking down larger TEP particles into smaller TEP precursors that would be mobilized to
the DOC pool and would thus maintain a relatively stable TEP pool although we observed a
positive increase in TEP with increased blooms of DDAs (see below section 3.4.1).

### 3.4 Production and utilization of TEP by primary and bacterial populations

Typically TEP are formed by diverse algal and bacterial species (Mari and Burd 1998)
yet are utilized mostly by bacteria and grazers as a rich C source (Engel and Passow, 2001;
Azam and Malfatti, 2007; Bar-Zeev et al., 2015). Throughout this experiment (P1 and P2
stages) TEP was not significantly correlated to parameters related to autotrophic production
such as total Chl $a$, PP, non-diazotrophic diatom or cyanobacterial abundance, or the growth
and mortality rates of these populations (Table S2). Furthermore, during P1, no significant
relationship between TEP and BA (total or specific for high and low nucleic acid bacteria-
HNA or LNA respectively), BP, or division rates was noted in any of the mesocosms (Table
S2).
This changed during P2 when TEP was positively correlated to the increasing BP for all
three mesocosms (Pearson's correlation coefficient $R^2$ = 0.63, 0.66, 0.69 for M1, M2, and M3
respectively, $p < 0.05$), (Fig. 5). During P2, TEP was also strongly and positively correlated to
TOC, which significantly increased over this time period (Fig. 4c) due to the high production
rates of both photosynthetic and heterotrophic bacterial populations. However, although BP
and PP were positively associated during P2 (log-log transformation, Fig. 5 in Van Wambeke
et al. this issue), we found no direct correlation between TEP and PP for either linear (Table
S2) or log-transformed regression (not shown). This coupling between PP and BP, while a
concurrent association between TEP and BP occurred during P2, indicates TEP may have been
utilized by bacteria as a carbon source (Azam, 1998; Ziervogel et al., 2014) or provided a
suitable niche for aggregation and proliferation of heterotrophic bacteria.

### 3.4.1 TEP and diazotrophic populations

Overall $N_2$ fixation rates were not significantly correlated with TEP concentrations at
any time in the experiment (Table S2). Neither could we discern any direct evidence of TEP
providing a carbon source for heterotrophic diazotrophs as was found previously in the Gulf
of Aqaba where these organisms contributed greatly to the $N_2$ fixation rates (Rahav et al.,



2015). Indeed, no relationship was found between TEP concentrations and the abundance or
growth rates of the heterotrophic diazotrophs γ-24774A11 (Moisander et al., 2014). Although
these organisms were present throughout the experiment, and increased ~4 fold from day 9 to
15 especially in M3, they contributed only a small fraction to the total diazotrophic biomass
and $N_2$ fixation rates (Turk-Kubo et al., 2015).

Yet, discerning individual diazotroph populations revealed some species-specific

correspondence to TEP at certain periods during the experiment. For example, throughout the
experiment, net growth rates (i.e., based on differences of *nifH* copies $L^{-1}$ from day to day) of
the DDA *Richelia* (Het-1) associated with *Rhizosolenia* (Turk-Kubo et al., 2015) temporally
paralleled TEP concentrations in all mesocosms (Fig. 6a-c, Fig. 6e-f). During both P1 and P2
TEP concentrations were positively correlated with the net growth rates of Het-1 ($R^2$=0.6
P=0.0001, n=19 for all mesocosms (Fig. 6d). Although the DDAs dominated the diazotroph
community during P1 (primarily Het-1), their overall contribution to diatom biomass in the
mesocosm was low with only 2-8% of all diatom biomass (Leblanc et al., this issue). We did
not observe an overall relationship between TEP and total diatom biomass throughout
VAHINE although diatoms are well known for their TEP production especially when
nutrients are limiting and growth rates decline (Urbani et al., 2005; Fukao et al., 2010). Thus,
the positive association between TEP and the growth rates of Het-1 and not of the other
DDAs Het-2 and Het-3 is intriguing.

TEP was also associated with the growth rates of the unicellular UCYN-C diazotrophs

that bloomed during P2 and dominated the $N_2$ fixation rates or this period (Turk-Kubo et al.,
2015; Berthelot et al., 2015). During P2, UCYN-C net growth rates were positively correlated
with increasing TEP concentrations ($R^2$= 0.65, 0.83, 0.88 for M1, M2, M3 respectively, p <
0.05). Furthermore, UCYN-C formed large aggregates (100-500 µm) embedded in an organic
matrix possibly also comprised of TEP (Fig. 6g-h) and were predominantly responsible for
the enhanced export production (22.4 ± 5% of exported POC), (Knapp et al., This issue;
Bonnet et al., This issue-a). High TEP content was obtained from sediment traps on days 15
and 16 (Fig. S1), corresponding to the height of the UCYN-C bloom in the mesocosms (Turk-
Kubo et al., 2015) and substantiating the role of TEP in facilitating export flux in the New
Caledonia lagoon (Mari et al., 2007).

The diazotroph *Trichodesmium*, that can account for huge surface blooms in the New

Caledonia lagoons (Rodier and Le Borgne, 2008; Rodier and Le Borgne, 2010), did not bloom
or accumulate within the VAHINE mesocosms. Yet, on day 23 a dense surface accumulation





was sighted on the surface of the lagoon waters (Spungin et al., This issue). Frequent
sampling (every 2-4 h) over the subsequent two days yielded extremely high TEP
concentrations (> 800 µg GX $L^{-1}$) from this rapidly declining biomass (Spungin et al., This
issue) corresponding to previous work demonstrating high TEP concentrations in
*Trichodesmium* from the New Caledonian lagoon that are undergoing autocatalytic
programmed cell death (PCD), (Berman-Frank et al., 2004; Berman-Frank et al., 2007; Bar-
Zeev et al., 2013). We showed that nutrient stressed, PCD-induced *Trichodesmium* diverts
available carbon from growth processes to produce large amounts of TEP (Berman-Frank and
Dubinsky, 1999; Berman-Frank et al., 2007). The TEP produced combines with the decaying
biomass to form large particles and aggregates that sink downwards (Bar-Zeev et al., 2013).
Here, we could not quantify the flux of matter obtained after this ephemeral bloom crashed.
Yet, it is reasonable to assume that the high TEP content and the > 90% decline in biomass
over a 24 h period resulted in a large downward flux of TEP-cellular debris aggregates as we
had observed previously under laboratory experiments (Berman-Frank et al., 2007; Bar-Zeev
et al., 2013)**.**

## 4    Conclusions

Although physically separated from the surrounding lagoon, TEP formation and

breakdown was difficult to tease out in the VAHINE mesocosms where abiotic drivers
(turbulence, shear forces, chemical coagulation) and biotic processes (algal and bacterial
production and utilization) maintained an apparently constant pool of TEP within the TOC.
Total TEP content was generally stable throughout the experimental period of 23 days and
comprised ~28% of the TOC in the mesocosms and lagoon with uniform distribution in the
three sampled depths of the 15 m deep-water column.

TEP concentrations appeared to be impacted indirectly via changes in DIP availability

as it was biologically consumed in the mesocosms after fertilization. Thus, declining P
availability (low DIP, rapid $T_{DIP}$, and increased APA) was associated with higher TEP content
in all mesocosms. TEP concentrations were also positively associated with net growth rates of
two important diazotrophic groups: the DDA *Richelia-Rhizosolenia* (Fig. 6e-f), during P1 and
P2 (excluding days 21-23); and UCYN-C diazotrophs which bloomed during P2. High TEP
content in the sediment traps during the UCYN-C bloom indicates that TEP may have been
part of the organic matrix associated with the large aggregates of UCYN-C that were exported
to the sediment traps (Fig. 6g-h).



TEP may have also provided bacteria with a rich organic carbon source especially
during P2 when higher BP (stimulated by the higher PP) was positively correlated higher TEP
concentrations. High production of TEP also occurred in the lagoon water outside the
mesocosms on day 23 during the decline of a short-lived dense surface bloom of the
diazotrophic *Trichodesmium* (Spungin et al., This issue) . Our results emphasize the
complexities of the natural system and suggest that to understand the role of compounds such
as TEP, and their contribution to the DOC and POC pools, a wider perspective and
methodologies be undertaken to examine and characterize the different components of marine
gels (not only carbohydrate-based), (Verdugo, 2012; Bar-Zeev et al., 2015)

**Author contributions**
IBF conceived and designed the investigation of TEP dynamics within the VAHINE project.
TM, FVW, IBF, DS, and ER participated in the experiment and performed analyses of
samples and data, KTK analysed diazotrophic populations. IBF and DS wrote the manuscript
with contributions from all co-authors.

**Acknowledgements**
Many thanks to Sophie Bonnet who created, designed, and successfully executed the
VAHINE project .The participation of IBF, DS, and ER in the VAHINE experiment was
supported by the German-Israeli Research Foundation (GIF), project number 1133-13.8/2011
to IBF and through a collaborative grant to IBF and SB from MOST Israel and the High
Council for Science and Technology (HCST)-France. Funding for this research was provided
by the Agence Nationale de la Recherche (ANR starting grant VAHINE ANR-13-JS06-0002),
INSU-LEFE-CYBER program, GOPS, IRD and M.I.O The authors thank the captain and
crew of the R/V Alis; the SEOH divers service from the IRD research center of Noumea (E.
Folcher, B. Bourgeois and A. Renaud) and from the Observatoire Océanologique de
Villefranche-sur-mer (OOV, J.M. Grisoni), the technical service and support of the IRD
research center of Noumea. Thanks also to C. Guieu, F. Louis and J.M. Grisoni from OOV for
mesocosm design and deployment advice. Special thanks to H. Berthelot and all other
participants and PIs of the project for the joint efforts and for making their data available for





further analyses. This work is in partial fulfillment of the requirements for a PhD thesis for D.
Spungin at Bar Ilan University.

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





**Figure legends**
**Figure 1.** Temporal changes in transparent exopolymeric particle (TEP) concentrations (µg
GX L$^{-1}$) during the VAHINE mesocosm experiment. Data shown are from daily sampling of
three depths (1, 6, 12 m) in each mesocosm. Data was analyzed according to the characterized
phases of the experiment based on the diazotrophic communities that developed in the
mesocosms (Turk-Kubo et al., 2015) and biogeochemical characteristics (Bonnet et al., This
issue-a). **a.** Mesocosm 1 (M1) **b.** Mesocosm 2 (M2), **c.** Mesocsom 3 (M3), **d.** samples from
the lagoon waters outside of the mesocosms (O). Phases: P0= days 2-4, P1= days 5-14, P2=
days 15-23. Linear regressions (Pearson) of TEP for each of the phases are designated by a
solid line, only when significant. Pearson correlations coefficients and significant values ($p <$
0.05) are represented in bold in Table S1.
**Figure 2.** Total content of transparent exopolymeric particles (TEP) per mesocosm and in the
lagoon waters surrounding the mesocosms. The average amount in g GX mesocosm$^{-1}$ for the
two periods of the experiment after DIP fertilization was calculated from the total daily
amount based on concentrations measured at three depths and integrated for the specific
volume per mesocosm or for an equivalent volume of lagoon water. Averages are represented
in boxplots as a function of two different phases: P1 = days 5-14 and P2 = days 15-23. Red
(mesocosm 1 - M1), blue (mesocosm 2- M2), green (mesocosm - M3) and black (Outside
lagoon O). Straight lines within the boxes mark the median. No significant differences were
observed between the phases or between the three mesocsoms and the outside lagoon
(Kruskal-Wallis non-parametric analysis of variance; $p > 0.05$).
**Figure 3.** Relationships between the concentration of transparent exopolymeric particles
(TEP), (µg GX L$^{-1}$) and **a.** dissolved inorganic phosphorus DIP (µmol L$^{-1}$), **b.** turnover time of
DIP -T$_{DIP}$ (d) and **c.** alkaline phosphatase activity (APA), (nmol L$^{-1}$ h$^{-1}$) in the three
mesocosms (M1-red; M2-blue; M3-green) during phase 2 (days 15-23). For a and b Pearson
linear regressions yielded an R$^2$ = 0.54, n=23 (TEP/DIP) and an R$^2$=0.52, n=26 (TEP/T$_{DIP}$),
and for c. Log-transformed (log(TEP) / log(APA)) with R$^2$ 0.68, n=25. All correlations were
significant ($p < 0.05$). Error bars represent ± 1 standard deviation.
**Figure 4. a.** Temporal dynamics of TEP carbon concentrations (TEP-C, µM) in relationship
to the average total organic carbon (TOC), (µg L$^{-1}$), (thin black line) in the mesocosms (M1-
red dots, M2-blue dots, M3-green dots, and black dots- Outside waters (O). Black solid line
designates TEP-C averaged for the three mesocosms (thick black line). TEP-C was measured



from 6 m depths and calculated according to Engel (2000). **b.** Temporal changes in the
percent of TEP-C from TOC (%) in mesocsoms (green dots), and %TEP-C in the lagoon
waters (Out), (black dots). **c.** Relationship between TEP concentrations ($\mu$g GX L$^{-1}$) and TOC
($\mu$mole L$^{-1}$), during phase 2 (days 15-23) for Mesocosm 1 (M1, red dots), Mesocosm 2 (M2,
blue dots), Mesocosm 3 (M3, green dots). Significant correlations were observed (Pearson)
for all mesocosms. $R^2$ = 0.75- M1, 0.73-M2, and 0.58-M3 respectively, n=7-8, $p < 0.05$.
Allstatistics are detailed in Table S2.(p=0.05, n= 7-8). Error bars represent $\pm$ 1 standard
deviation.
**Figure 5.** Relationship between heterotrophic bacterial production (BP), (ng C L$^{-1}$ h$^{-1}$) and
TEP concentrations ($\mu$g GX L$^{-1}$) during phase 2 (days 15-23) when BP increased following
the enhanced PP (Van Wambeke et al., This issue), for Mesocosm 1 (M1, red dots),
Mesocosm 2 (M2, blue dots), Mesocosm 3 (M3, green dots). Pearson's linear regressions
yielded $R^2$ = 0.57 for M1, 0.42 for M2, and 0.56 for M3 respectively. Significant correlations
were observed for all mesocosms and are detailed in Table S2. Error bars represent $\pm$ 1
standard deviation.
**Figure 6.** Temporal changes in TEP concentrations and Het-1 net growth rates (d$^{-1}$), (gray
triangles) for **a.** Mesocosm 1 (M1) **b.** Mesocosm 2 (M2), **c.** Mesocsom 3 (M3). TEP
concentrations were averaged from the three depths sampled per mesocosm (green circles).
Het-1 net growth rates were calculated based on changes of *nifH* copies L$^{-1}$ (Turk-Kubo et al.,
2015) measured every other day. **d.** Relationship between TEP concentrations ($\mu$g GX L$^{-1}$)
and
Het-1 growth rate (d$^{-1}$) for all three mesocosms. Significant correlations were observed
(Pearson) from all mesocosms together. $R^2$ = 0.60, p=0.0001, n=19. Error bars represent $\pm$ 1
standard deviation. **e-f.** Epifluorescent microscopical images of the diatom-diazotroph
association *Richelia-Rhizosolenia* identified by Het-1 abundance. Images by V. Cornet-
Barthaux. **g-h.** the diazotroph UCYN-C which bloomed and formed large aggregates
(comprised also of TEP) that enhanced vertical flux and export production during P2. Images
by S. Bonnet.





**Figure 1**

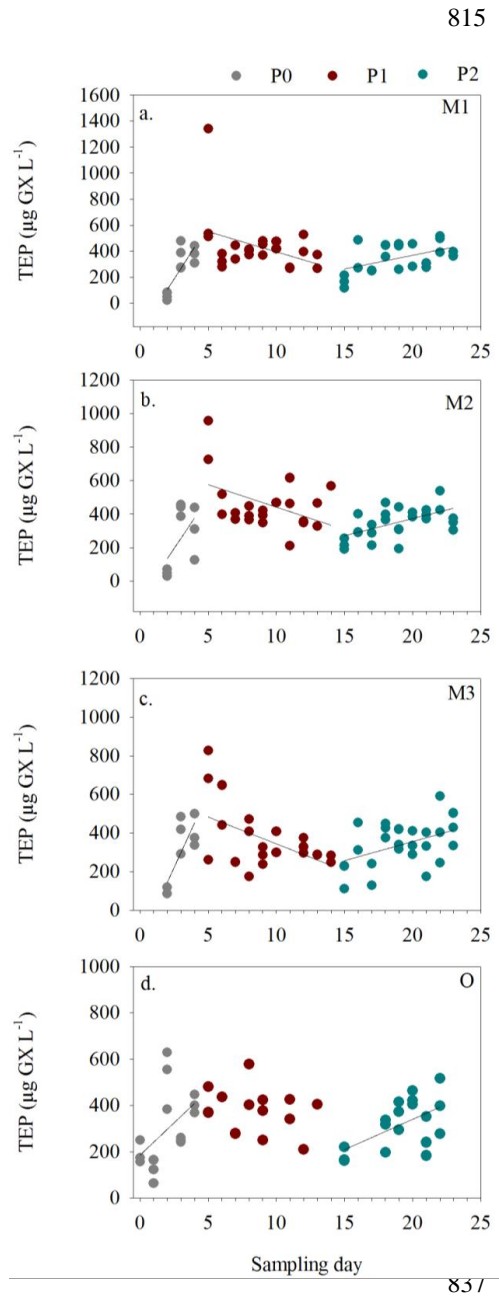






**Figure 2**















**Figure 3**


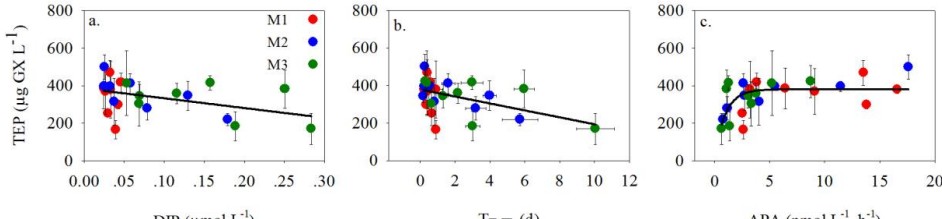












**Figure 4**


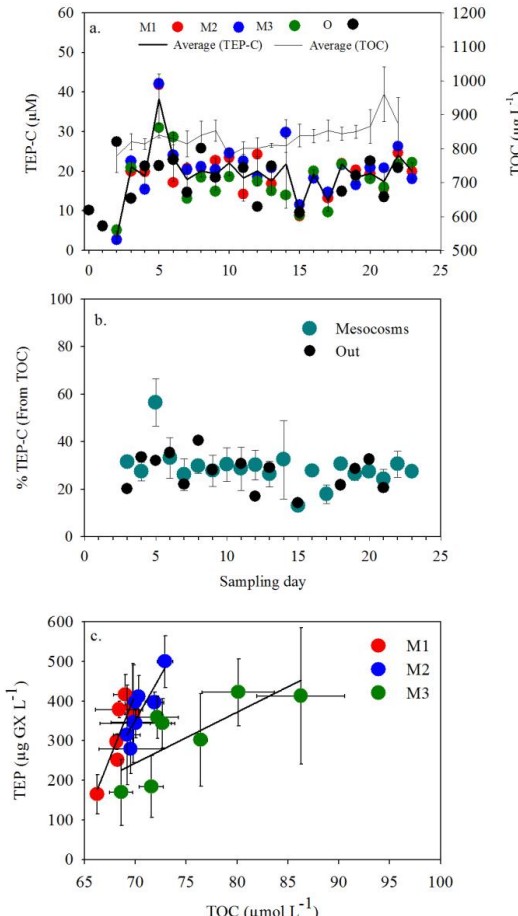








**Figure 5**


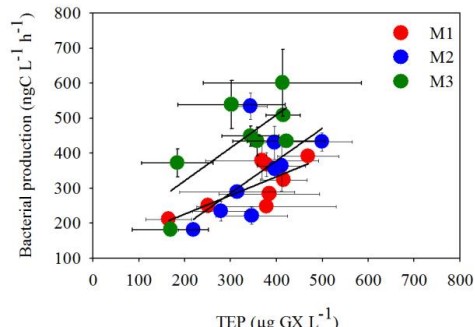







**Figure 6**


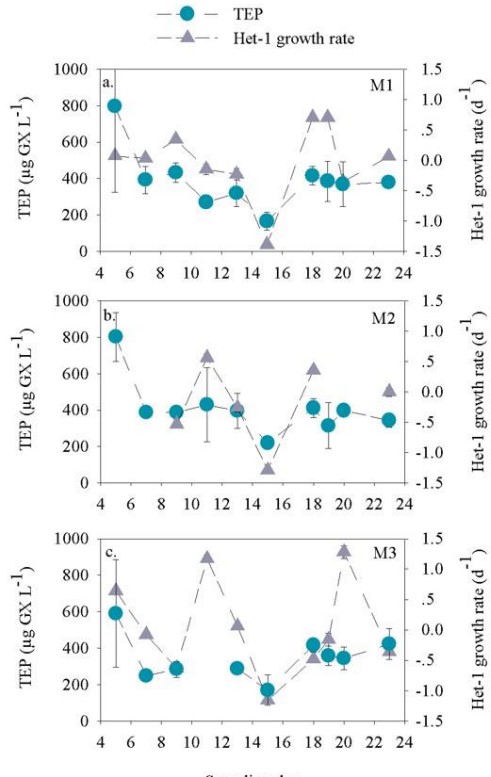

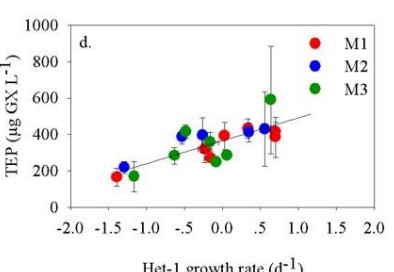

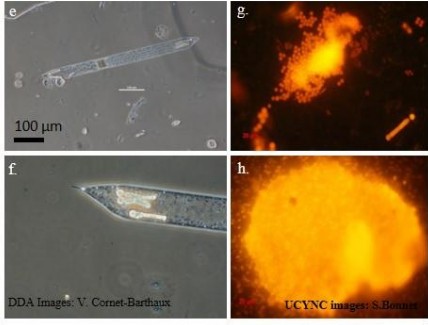





