# Peer review of "Dynamics of transparent exopolymer particles (TEP) during the VAHINE mesocosm experiment in the New Caledonia lagoon"

_Biogeosciences, 2015_

## Referee Comment (RC1) · Anonymous Referee #1 · 24 Feb 2016

This is a straightforward study with rather clear results showing that the TEP pool in the VAHINE experiment was fairly consistent and not greatly affected by the addition of phosphate or the reduction in currents caused by the mesocosms themselves. This is an interesting (although admittedly frustrating) result. I would like to see a little more discussion of the fact that the lagoon sample showed very similar temporal dynamics to the mesocosms, although not as pronounced. What might have been happening there? The researchers did a good job with methodology and interpretation. The interpretation of results are hampered by the lack of methods to specifically look at production and consumption of the TEP pool, but this is acknowledged by the authors and not within their control. Specific comments are below:

[Figure]

Line 128: I think you mean microM, rather than micromol.

Line 134-135: describe briefly here the rationale for the delineation of the P1 and P2 time periods. I know they are described in other parts of this special issue, but if someone were to read only this paper, it would be good to describe why this choice was made here.

Line 230: close parentheses around Synechococcus

Line 274-277: I think the reader would benefit from a slightly different description of the trends seen in the first days. Upon my first reading, I only imagined the spike that occurred after the phosphate addition, but the TEP was increasing during the entire P0 phase, spiked in the hours after phosphate addition, then decreased during P1

Line 284-285: the sections seem to be mixed up here. Do you mean that the lagoon increased in TEP during P0 and P2, but decreased during P1?

Line 350-351: DIP turnover rates indicate DIP stress or deficiency. That cannot fully indicate limitation without some sort of calibration

Line 353: meaning that TDIP needs to be >2d?

Line 467: I suspect the organic matrix around the UCYN-C was EPS produced by and remaining close to the cells (similar to what some phenotypes of UCYN-B do), rather than material that was released and then aggregated free-living cells of UCYN-C. I know it's a small distinction, and perhaps meaningless to many, but I also think it's worth noting that this is a possible scenario and there is precedent to believe that is what happened.

Lines 473-490: was this Trichodesmium bloom at the lagoon control sampling site, or elsewhere in the lagoon? Does it explain any of the results from the experiment or the lagoon results? If not, I don't really think it belongs here, as it is a description of a non-related phenomenon.

Figure 1: I would like to see all the figures put on the same X axis to make them more directly comparable. I know it will be harder to see patterns, but the comparison is more important, I think

[Figure]

---

## Referee Comment (RC2) · Anonymous Referee #2 · 9 Mar 2016

It is one of a series of papers describing long and extensive large-scale bioassay experiment involving many researchers. The multinational team has accumulated an extensive dataset that is presented as a series of papers in the special issue of Biogeosciences. Unfortunately, the mesocosm manipulation did not induce expected (and measurable) change in TEP concentration. Instead, TEP production and consumption seem to be tightly coregulated. Despite this almost negative outcome, the authors managed to prepare well written report that uses the wealth of available data for detailed analysis of the trends and correlations. The unique dataset obtained during the experiment is worth publishing, even if it does not bring any major breakthroughs. It proves the complexity of the microbial system in the lagoon and its ability to maintain

homeostasis. I have the following questions: 1. Looking at Fig.1, it seems that the only major difference between the mesocosms and the surrounding water is the spike in TEP concentration inside mesocosms immediately after the P addition. The other trends seem to be similar. Any idea why? 2. Did the authors check whether the optical absorption method using the Alcian blue dye staining to determine TEP concentration is linear? Was the filter absorption measured using the integrating sphere? If not, was there any significant scattering? 3. Do the authors have control data from the lagoon outside the mesocosms to be added into Figs.3-6? 4. Any idea why was TOC significantly higher in M3? Why TEP did not increase proportionally?

---

## Author Comment (AC2) · 2 Apr 2016

Reviewer 2 1. Looking at Fig.1, it seems that the only major difference between the mesocosms and the surrounding water is the spike in TEP concentration inside mesocosms immediately after the P addition. The other trends seem to be similar. Any idea why?

Author: This has been described in section 3.1.1 . The difference between the TEP in the mesocosms and the lagoon water is hard to see and is significantly different immediately after P addition and only during P1 after P addition and subsequent utilization when declining P availability was correlated with increased TEP concentrations. The decline in TEP concentrations from the lagoon water during P1 was not statistically significant as demonstrated in the mesocosms (Fig. 1, Fig. S1). The significant decline in TEP in the first days after P addition is probably due to two factors: a) phytoplankton relieved of P stress will produce less TEP and increase growth rates, b) bacteria will utilize the added P as well as TEP and other organic C sources to grow – so higher TEP consumption and therefore a more significant decline in the mesos compared to the lagoon.

- 2. Did the authors check whether the optical absorption method using the Alcian blue dye staining to determine TEP concentration is linear? Was the filter absorption measured using the integrating sphere? If not, was there any significant scattering? Author:TEP concentrations are determined from an Alcian blue (AB) calibration curve done. AB was calibrated using different volumes of purified polysaccharide GX and - The absorption measured was done with a spectrophotometer (Cary 100) equipped with an integrated sphere.

3. Do the authors have control data from the lagoon outside the mesocosms to be added into Figs.3-6?

Author:Control data for figures 3-5 are available in supplementary figures we show here (in the supplementary pdf). As none of them had any significant correlation we decided not to show them but only state this in the text.For Fig 6 no data exists of DDA growth rates in the lagoon (control) water. We have also added all the statistics we performed for the control (out) versus the parameters tested for the mesocosms in the revised supplementary Table S2 (attached here in supplemental pdf)

4. Any idea why was TOC significantly higher in M3? Why TEP did not increase proportionally?

Author: M3 had higher biomass both PP and bacterial which enriched TOC (Berthelot et al. 2015 ) and a full discussion on the replicability and variability of the mesocosms can be found in the introductory paper to the project (Bonnet et al. 2016) . Why TEP did not increase proportionally is a good question – although when we look at fig 5 we

can see a similar slope of BP to TEP concentration but shifted to higher production levels of BP that were found in M3. The higher BP possibly indicates a greater extent of utilization of TEP and organic C so that the resulting concentrations which we measured did not significantly change.

Please also note the supplement to this comment:
http://www.biogeosciences-discuss.net/bg-2015-612/bg-2015-612-AC2-supplement.pdf

**Supplement:**

April 2, 2016

Dear Editor.

We thank the reviewers for their time and suggestions. Attached please find our answers to the specific comments raised by the reviewers and some extra figures and supplementary table we have added. **Our replies are in bold**

❑Our replies

Reviewer 1

1.Line 128: I think you mean microM, rather than micromol.

**Yes, changed**

2.Line 134-135: describe briefly here the rationale for the delineation of the P1 and P2 time periods. I know they are described in other parts of this special issue, but if someone were to read only this paper, it would be good to describe why this choice was made here.

**The chemical and biological changes that occurred in each of the experimental stages are described in detail at the beginning of the results section. Section 3.1). We have, however, added the following sentences at end of section 2.1 to clarify the rationale of these delineations**.

"Based on the results of different biogeochemical and biological parameters during VAHINE (Turk-Kubo et al., 2015; Berthelot et al., 2015; Bonnet et al., ), three specific periods were discerned (see detailed description in section 3.1) within which we have also investigated TEP dynamics: Days 2-4 (P0) are the pre-fertilization days when the DIP concentrations were 0.02-0.05 $PO_4^{3-}$ and combined DIN were extremely low; days 5-14 (P1) –After fertilization on day 5 the $PO_4^{3-}$ concentrations were ~0.8 µmol $L^{-1}$ and diazotrophic populations were dominated by diatom-diazotroph associations. The second stage of the experiment (P2) from days 15 to 23 was characterized by simultaneous increase in primary and bacterial production as well as in $N_2$ fixation rates which averaged 27.7 nmol $L^{-1}$ $d^{-1}$ (Berthelot et al. 2015) and diazotrophic populations comprised primarily by the unicellular UCYN-C (Turk-kubo et al 2015)."

3.Line 230: close parentheses around Synechococcus

**parentheses added**

4.Line 274-277: I think the reader would benefit from a slightly different description of the trends seen in the first days. Upon my first reading, I only imagined the spike that occurred after the phosphate addition, but the TEP was increasing during the entire P0 phase, spiked in the hours after phosphate addition, then decreased during P1

**Line 274-277: TEP concentrations increased from the lowest volumetric concentrations (averaging~ 50 µg GX L-1) measured on day 2 to reach maximum concentrations in each of the mesocosms (average of ~800 µg GX L-1) on day 5, ~15 h after the mesocosms were fertilized with DIP (Fig. S1, Fig. 1a).**

5.Line 284-285: the sections seem to be mixed up here. Do you mean that the lagoon increased in TEP during P0 and P2, but decreased during P1?

**Yes, there was a mistake in the sentence. Fixed to:**

**"TEP concentrations in the lagoon water were compared with those in the mesocosms. These showed a similar pattern of increase in TEP during P0 and P2 while the gradual decline in TEP concentrations during P1 was not statistically significant as observed in the mesocosms (Fig. 1, Fig. S1)."(there is no P3 it's a mistake- deleted and change in text)**

6.Line 350-351: DIP turnover rates indicate DIP stress or deficiency. That cannot fully indicate limitation without some sort of calibration.

**You are right. Turnover rates alone do not indicate deficiency. However, increasing Alkaline phosphatase activity (APA) in M1 and M2 from day 18 and in M3 from day 21 suggests that the cells were responding to P stress (Van Wambeke et al. this issue). We have rephrased the sentence**

7. Line 467: I suspect the organic matrix around the UCYN-C was EPS produced by and remaining close to the cells (similar to what some phenotypes of UCYN-B do), rather than material that was released and then aggregated free-living cells of UCYN-C. I know it's a small distinction, and perhaps meaningless to many, but I also think it's worth noting that this is a possible scenario and there is precedent to believe that is what happened.

**We agree with you as to the mechanism of aggregation. We have modified the sentence making this distinction**

**"Furthermore, UCYN-C probably produced an organic matrix possibly also comprised of TEP that aided the formation of large aggregates (100-500 µm) (Fig. 6g-h). These aggregates were predominantly responsible for the enhanced export production (22.4 ± 5% of exported POC), (Knapp et al., 2015; Bonnet et al., 2016 – both in this special issue-."**

8.Lines 473-490: was this Trichodesmium bloom at the lagoon control sampling site, or elsewhere in the lagoon? Does it explain any of the results from the experiment or the lagoon results? If not, I don't really think it belongs here, as it is a description of a non-related phenomenon.

**Yes, Trichodesmium bloomed at the lagoon control sampling site. However, upon rereading the paragraph and your comment, we agree that it does not provide any further explanation of the results and have thus removed the whole paragraph.**

Paper Figure 1: I would like to see all the figures put on the same X axis to make them more directly comparable. I know it will be harder to see patterns, but the comparison is more important, I think

**All plots of Fig 1 are on the same X axis. We have now presented them with the same Y scale. We here include a supplemental figure with the average TEP concentration from each mesocosm and from the lagoon on the same plot to easily compare. These show how uniform overall TEP content is but when dissected each mesocosm shows a similar pattern of increase and decrease that we want to emphasize.**

[Figure]

Reviewer 2

1. Looking at Fig.1, it seems that the only major difference between the mesocosms and the surrounding water is the spike in TEP concentration inside mesocosms immediately after the P addition. The other trends seem to be similar. Any idea why?

**This has been described in section 3.1.1 . The difference between the TEP in the mesocosms and the lagoon water is hard to see and is significantly different immediately after P addition and only during P1 after P addition and subsequent utilization when declining P availability was correlated with increased TEP concentrations. The decline in TEP concentrations from the lagoon water during P1 was not statistically significant as demonstrated in the mesocosms (Fig. 1, Fig. S1). The significant decline in TEP in the first days after P addition is probably due to two factors: a) phytoplankton relieved of P stress will produce less TEP and increase growth rates, b) bacteria will utilize the added P as well as TEP and other organic C sources to grow – so higher TEP consumption and therefore a more significant decline in the mesos compared to the lagoon.**

- 2. Did the authors check whether the optical absorption method using the Alcian blue dye staining to determine TEP concentration is linear? Was the filter absorption measured using the integrating sphere? If not, was there any significant scattering?

**TEP concentrations are determined from an Alceline blue (AB) calibration curve done. AB was calibrated using different volumes of purified polysaccharide GX and - The absorption measured was done with a spectrophotometer (Cary 100) equipped with an integrated sphere.**

3. Do the authors have control data from the lagoon outside the mesocosms to be added into Figs.3-6?

**Control data for figures 3-5 are available in supplementary figures we show here (below). As none of them had any significant correlation we decided not to show them but only state this in the text.**

**For Fig 6 no data exists of DDA growth rates in the lagoon (control) water. We have also added all the statistics we performed for the control (out) versus the parameters tested for the mesocosms in the revised supplementary Table S2 (attached here)**

4. Any idea why was TOC significantly higher in M3? Why TEP did not increase proportionally?

M3 had higher biomass both PP and bacterial which enriched TOC (Berthelot et al. 2015 ) and a full discussion on the replicability and variability of the mesocosms can be found in the introductory paper to the project (Bonnet et al. 2016) . Why TEP did not increase proportionally is a good question – although when we look at fig 5 we can see a similar slope of BP to TEP concentration but shifted to higher production levels of BP that were found in M3.  The higher BP possibly indicates a greater extent of utilization of TEP and organic C so that the resulting concentrations which we measured did not significantly change.

[Figure]

**Figure 3S – relationship between TEP concentrations to DIP, TDIP, and APA activity measured in the lagoon throughout the VAHINE experiment.**

[Figure]

**Figure 4S. Relationship between TEP concentrations measured in the Lagoon (outside the mesocosms) to concentrations of total organic carbon ( TOC) and bacterial production measured throughout the VAHINE experiment.**

**Table S2.** Pearson's linear regression analyses between the average concentration of transparent exopolymeric particles [TEP (µg GX L-1 )] and the physical, chemical, and biological parameters from each mesocosm (M1, M2, M3) and outside waters (O) divided into the two postfertilization phases of the VAHINE experiment. P1 = days 5-14, P2 = days 15-23. Each TEP value is an average of the measurements from three sampled depths. Correlations in bold are statistically significant with P < 0.05. For Het-1 and UCYN-C the growth rate (µ) is the net growth rate, based on changes of nifH copies L-1 from day to day.

| Parameter | Mesocosm | Period | R2 | P | n |
|---|---|---|---|---|---|
| | M1 | | 0.055 | 0.577 | 8 |
| | M2 | | 0.015 | 0.776 | 8 |
| | | P1 | | | |
| | M3 | | 0.191 | 0.279 | 8 |
| Temperature (°C) | O | | 0.156 | 0.381 | 7 |
| | M1 | | 0.369 | 0.148 | 7 |
| | M2 | | 0.087 | 0.520 | 7 |
| | | P2 | | | |
| | M3 | | 0.357 | 0.157 | 7 |
| | O | | 0.001 | 0.955 | 5 |
| | M1 | | 0.011 | 0.805 | 8 |
| | M2 | | 0.055 | 0.544 | 9 |
| | | P1 | | | |
| | M3 | | 0.295 | 0.163 | 8 |
| DIP (µmol L$^{-1}$) | O | | 0.038 | 0.677 | 7 |
| | M1 | | 0.031 | 0.676 | 8 |
| | M2 | | 0.539 | **0.038** | 8 |
| | | P2 | | | |
| | M3 | | 0.249 | 0.123 | 8 |
| | O | | 0.171 | 0.415 | 6 |
| | M1 | | 0.000 | 0.965 | 8 |
| | M2 | | 0.198 | 0.229 | 9 |
| | | P1 | | | |
| | M3 | | 0.004 | 0.879 | 8 |
| DOP (µmol L$^{-1}$) | O | | 0.042 | 0.658 | 7 |
| | M1 | | 0.128 | 0.383 | 8 |
| | M2 | | 0.367 | 0.112 | 8 |
| | | P2 | | | |
| | M3 | | 0.141 | 0.320 | 9 |
| | O | | 0.705 | 0.075 | 5 |
| | M1 | | 0.020 | 0.738 | 8 |
| | M2 | | 0.050 | 0.563 | 9 |
| | | P1 | | | |
| | M3 | | 0.039 | 0.641 | 8 |
| POP (µmol L$^{-1}$) | O | | 0.145 | 0.399 | 7 |
| | M1 | | 0.103 | 0.401 | 9 |
| | M2 | | 0.005 | 0.851 | 9 |
| | | P2 | | | |
| | M3 | | 0.192 | 0.237 | 9 |
| | O | | 0.0005 | 0.968 | 6 |

| | | | | | |
|---|---|---|---|---|---|
| | M1 | | 0.077 | 0.51 | 8 |
| | M2 | P1 | 0.012 | 0.775 | 9 |
| | M3 | | 0.043 | 0.620 | 8 |
| $T_{DIP}$ (d) | O | | 0.073 | 0.557 | 7 |
| | M1 | | 0.238 | 0.182 | 9 |
| | M2 | P2 | 0.523 | **0.028** | 9 |
| | M3 | | 0.338 | 0.100 | 9 |
| | O | | 0.239 | 0.325 | 6 |
| | M1 | | 0.155 | 0.294 | 9 |
| | M2 | P1 | 0.432 | **0.077** | 8 |
| | M3 | | 0.048 | 0.638 | 7 |
| APA | O | | 0.075 | 0.553 | 7 |
| (nmole $L^{-1}$ $h^{-1}$) | M1 | | 0.173 | 0.265 | 9 |
| | M2 | P2 | 0.683 | **0.011** | 8 |
| | M3 | | 0.300 | 0.126 | 9 |
| | O | | 0.281 | 0.280 | 6 |
| | M1 | | 0.005 | 0.879 | 7 |
| | M2 | P1 | 0.003 | 0.882 | 9 |
| | M3 | | 0.051 | 0.591 | 8 |
| DOC | O | | 0.036 | 0.686 | 7 |
| (µmol $L^{-1}$) | M1 | | 0.266 | 0.295 | 6 |
| | M2 | P2 | 0.268 | 0.482 | 4 |
| | M3 | | 0.285 | 0.275 | 6 |
| | O | | 0.008 | 0.888 | 5 |
| | M1 | | 0.213 | 0.211 | 9 |
| | M2 | P1 | 0.005 | 0.853 | 9 |
| | M3 | | 0.216 | 0.246 | 8 |
| POC | O | | 0.099 | 0.493 | 7 |
| (µmol $L^{-1}$) | M1 | | 0.006 | 0.883 | 6 |
| | M2 | P2 | 0.212 | 0.358 | 6 |
| | M3 | | 0.911 | **0.046** | 4 |
| | O | | 0.014 | 0.883 | 4 |
| | M1 | | 0.105 | 0.434 | 8 |
| TOC | M2 | P1 | 0.003 | 0.883 | 9 |
| (µmol $L^{-1}$) | M3 | | 0.002 | 0.926 | 8 |
| | O | | 0.006 | 0.869 | 7 |

| | | | | | |
|---|---|---|---|---|---|
| | M1 | | 0.745 | 0.012 | 7 |
| | M2 | P2 | 0.728 | 0.007 | 8 |
| | M3 | | 0.582 | 0.046 | 7 |
| | O | | 0.222 | 0.422 | 5 |
| | M1 | | 0.112 | 0.417 | 8 |
| | M2 | P1 | 0.042 | 0.597 | 9 |
| | M3 | | 0.041 | 0.632 | 8 |
| DON | O | | 0.037 | 0.677 | 7 |
| (µmol L⁻¹) | M1 | | 0.166 | 0.316 | 8 |
| | M2 | P2 | 0.718 | **0.008** | 8 |
| | M3 | | 0.379 | 0.104 | 8 |
| | O | | 0.061 | 0.638 | 6 |
| | M1 | | 0.381 | 0.103 | 8 |
| | M2 | P1 | 0.160 | 0.286 | 9 |
| | M3 | | 0.334 | 0.133 | 8 |
| PON | O | | 0.084 | 0.527 | 7 |
| (µmol L⁻¹) | M1 | | 0.000 | 0.990 | 6 |
| | M2 | P2 | 0.330 | 0.233 | 6 |
| | M3 | | 0.036 | 0.720 | 6 |
| | O | | 0.232 | 0.519 | 4 |
| | M1 | | 0.041 | 0.629 | 8 |
| | M2 | P1 | 0.325 | 0.140 | 8 |
| | M3 | | 0.007 | 0.858 | 7 |
| N₂ fixation | O | | 0.0002 | 0.980 | 6 |
| (nmol L⁻¹ d⁻¹) | M1 | | 0.046 | 0.579 | 9 |
| | M2 | P2 | 0.038 | 0.617 | 9 |
| | M3 | | 0.405 | 0.065 | 9 |
| | O | | 0.267 | 0.293 | 6 |
| | M1 | | 0.251 | 0.169 | 9 |
| | M2 | P1 | 0.080 | 0.460 | 9 |
| | M3 | | 0.054 | 0.581 | 8 |
| Chlorophyll a | O | | 0.056 | 0.609 | 7 |
| (µg L⁻¹) | M1 | | 0.096 | 0.418 | 9 |
| | M2 | P2 | 0.126 | 0.348 | 9 |
| | M3 | | 0.292 | 0.133 | 9 |
| | O | | 0.057 | 0.649 | 6 |

Note: The $N_2$ fixation values use $\mu$ notation for micro where applicable; DON and PON are in $\mu mol\ L^{-1}$, N₂ fixation in $nmol\ L^{-1}\ d^{-1}$, Chlorophyll a in $\mu g\ L^{-1}$.

| | | | | | |
|---|---|---|---|---|---|
| | M1 | | 0.078 | 0.504 | 8 |
| | M2 | | 0.046 | 0.577 | 9 |
| | M3 | P1 | 0.209 | 0.254 | 8 |
| PP | O | | 0.029 | 0.713 | 7 |
| ($\mu$mol C L$^{-1}$ d$^{-1}$) | M1 | | 0.000 | 0.991 | 8 |
| | M2 | | 0.332 | 0.105 | 9 |
| | M3 | P2 | 0.124 | 0.392 | 8 |
| | O | | 0.499 | 0.117 | 6 |
| | M1 | | 0.083 | 0.488 | 8 |
| | M2 | | 0.000 | 0.973 | 9 |
| | M3 | P1 | 0.549 | **0.035** | 8 |
| BB | O | | 0.266 | 0.236 | 7 |
| (ngC L$^{-1}$ h$^{-1}$) | M1 | | 0.574 | **0.029** | 8 |
| | M2 | | 0.424 | **0.058** | 9 |
| | M3 | P2 | 0.567 | **0.031** | 8 |
| | O | | 0.153 | 0.444 | 6 |
| | M1 | P1 | 0.767 | 0.124 | 4 |
| | M2 | | 0.999 | **0.021** | 3 |
| | M3 | | N.A | N.A | N.A |
| Het1 | O | | - | - | - |
| ($\mu$) | M1 | P2 | 0.837 | **0.029** | 5 |
| | M2 | | 0.754 | 0.132 | 4 |
| | M3 | | 0.137 | 0.540 | 5 |
| | O | | - | - | - |
| | M1 | P1 | N.A | N.A | N.A |
| | M2 | | 0.005 | 0.953 | 3 |
| | M3 | | N.A | N.A | N.A |
| UCYN-C | O | | - | - | - |
| ($\mu$) | M1 | P2 | 0.421 | 0.236 | 5 |
| | M2 | | 0.694 | 0.167 | 4 |
| | M3 | | 0.775 | **0.049** | 5 |
| | O | | - | **-** | - |

DIP: dissolved inorganic *phosphate*; DOP and POP: dissolved and particulate organic *phosphate*; T$_{DIP}$: Turnover rates of dissolved inorganic phosphate; APA: Alkaline phosphatase activity; DOC and POC: dissolved and particulate organic carbon; TOC: total organic carbon; DON and PON: dissolved and particulate organic nitrogen; BP and PP- bacterial and primary production.

---

## Author Response (AR1)

April 2, 2016

**Dear Editor.**

We thank the reviewers for their time and suggestions. Attached please find our answers to the specific comments raised by the reviewers and some extra figures and supplementary table we have added. **Our replies are in bold**

Dur replies

Reviewer 1

1.Line 128: I think you mean microM, rather than micromol.

**Yes, changed**

2.Line 134-135: describe briefly here the rationale for the delineation of the P1 and P2 time periods. I know they are described in other parts of this special issue, but if someone were to read only this paper, it would be good to describe why this choice was made here.

**The chemical and biological changes that occurred in each of the experimental stages are described in detail at the beginning of the results section. Section 3.1). We have, however, added the following sentences at end of section 2.1 to clarify the rationale of these delineations.**

"Based on the results of different biogeochemical and biological parameters during VAHINE (Turk-Kubo et al., 2015; Berthelot et al., 2015; Bonnet et al., ), three specific periods were discerned (see detailed description in section 3.1) within which we have also investigated TEP dynamics: Days 2-4 (P0) are the pre-fertilization days when the DIP concentrations were  $0.02-0.05 \text{ PO}_4^{3-}$  and combined DIN were extremely low; days 5-14 (P1) –After fertilization on day 5 the PO43- concentrations were ~0.8 µmol L-1 and diazotrophic populations were dominated by diatom-diazotroph associations. The second stage of the experiment (P2) from days 15 to 23 was characterized by simultaneous increase in primary and bacterial production as well as in N2 fixation rates which averaged 27.7 nmol L-1 d-1 (Berthelot et al. 2015) and diazotrophic populations comprised primarily by the unicellular UCYN-C (Turk-kubo et al 2015)."

3.Line 230: close parentheses around Synechococcus

**parentheses added**

4.Line 274-277: I think the reader would benefit from a slightly different description of the trends seen in the first days. Upon my first reading, I only imagined the spike that occurred after the phosphate addition, but the TEP was increasing during the entire PO phase, spiked in the hours after phosphate addition, then decreased during P1

Line 274-277: TEP concentrations increased from the lowest volumetric concentrations (averaging  $\sim$  50 µg GX L-1) measured on day 2 to reach maximum concentrations in each of the mesocosms (average of ~800 µg GX L-1) on day 5, ~15 h after the mesocosms were fertilized with DIP (Fig. S1, Fig. 1a).

5.Line 284-285: the sections seem to be mixed up here. Do you mean that the lagoon increased in TEP during P0 and P2, but decreased during P1?

Yes, there was a mistake in the sentence. Fixed to:

"TEP concentrations in the lagoon water were compared with those in the mesocosms. These showed a similar pattern of increase in TEP during P0 and P2 while the gradual decline in TEP concentrations during P1 was not statistically significant as observed in the mesocosms (Fig. 1, Fig. S1)." (there is no P3 it's a mistake- deleted and change in text)

6.Line 350-351: DIP turnover rates indicate DIP stress or deficiency. That cannot fully indicate limitation without some sort of calibration.

You are right. Turnover rates alone do not indicate deficiency. However, increasing Alkaline phosphatase activity (APA) in M1 and M2 from day 18 and in M3 from day 21 suggests that the cells were responding to P stress (Van Wambeke et al. this issue). We have rephrased the sentence

7. Line 467: I suspect the organic matrix around the UCYN-C was EPS produced by and remaining close to the cells (similar to what some phenotypes of UCYN-B do), rather than material that was released and then aggregated free-living cells of UCYN-C. I know it's a small distinction, and perhaps meaningless to many, but I also think it's worth noting that this is a possible scenario and there is precedent to believe that is what happened.

We agree with you as to the mechanism of aggregation. We have modified the sentence making this distinction

"Furthermore, UCYN-C probably produced an organic matrix possibly also comprised of TEP that aided the formation of large aggregates (100-500  $\mu$ m) (Fig. 6g-h). These aggregates were predominantly responsible for the enhanced export production (22.4 ± 5% of exported POC), (Knapp et al., 2015; Bonnet et al., 2016 – both in this special issue-."

8.Lines 473-490: was this Trichodesmium bloom at the lagoon control sampling site, or elsewhere in the lagoon? Does it explain any of the results from the experiment or the lagoon results? If not, I don't really think it belongs here, as it is a description of a non-related phenomenon.

Yes, Trichodesmium bloomed at the lagoon control sampling site. However, upon rereading the paragraph and your comment, we agree that it does not provide any further explanation of the results and have thus removed the whole paragraph.

Paper Figure 1: I would like to see all the figures put on the same X axis to make them more directly comparable. I know it will be harder to see patterns, but the comparison is more important, I think

All plots of Fig 1 are on the same X axis. We have now presented them with the same Y scale. We here include a supplemental figure with the average TEP concentration from each mesocosm and from the lagoon on the same plot to easily compare. These show how uniform overall TEP content is but when dissected each mesocosm shows a similar pattern of increase and decrease that we want to emphasize.

**Reviewer 2**

1. Looking at Fig.1, it seems that the only major difference between the mesocosms and the surrounding water is the spike in TEP concentration inside mesocosms immediately after the P addition. The other trends seem to be similar. Any idea why?

This has been described in section 3.1.1. The difference between the TEP in the mesocosms and the lagoon water is hard to see and is significantly different immediately after P addition and only during P1 after P addition and subsequent utilization when declining P availability was correlated with increased TEP concentrations. The decline in TEP concentrations from the lagoon water during P1 was not statistically significant as demonstrated in the mesocosms (Fig. 1, Fig. S1). The significant decline in TEP in the first days after P addition is probably due to two factors: a) phytoplankton relieved of P stress will produce less TEP and increase growth rates, b) bacteria will utilize the added P as well as TEP and other organic C sources to grow – so higher TEP consumption and therefore a more significant decline in the mesos compared to the lagoon.

- 2. Did the authors check whether the optical absorption method using the Alcian blue dye staining to determine TEP concentration is linear? Was the filter absorption measured using the integrating sphere? If not, was there any significant scattering?

TEP concentrations are determined from an Alceline blue (AB) calibration curve done. AB was calibrated using different volumes of purified polysaccharide GX and - The absorption measured was done with a spectrophotometer (Cary 100) equipped with an integrated sphere.

3. Do the authors have control data from the lagoon outside the mesocosms to be added into Figs.3-6?

Control data for figures 3-5 are available in supplementary figures we show here (below). As none of them had any significant correlation we decided not to show them but only state this in the text.

For Fig 6 no data exists of DDA growth rates in the lagoon (control) water. We have also added all the statistics we performed for the control (out) versus the parameters tested for the mesocosms in the revised supplementary Table S2 (attached here)

4. Any idea why was TOC significantly higher in M3? Why TEP did not increase proportionally?

M3 had higher biomass both PP and bacterial which enriched TOC (Berthelot et al. 2015) and a full discussion on the replicability and variability of the mesocosms can be found in the introductory paper to the project (Bonnet et al. 2016). Why TEP did not increase proportionally is a good question – although when we look at fig 5 we can see a similar slope of BP to TEP concentration but shifted to higher production levels of BP that were found in M3. The higher BP possibly indicates a greater extent of utilization of TEP and organic C so that the resulting concentrations which we measured did not significantly change.

Figure 3S – relationship between TEP concentrations to DIP, TDIP, and APA activity measured in the lagoon throughout the VAHINE experiment.

Figure 4S. Relationship between TEP concentrations measured in the Lagoon (outside the mesocosms) to concentrations of total organic carbon (TOC) and bacterial production measured throughout the VAHINE experiment.

**Table S2.** Pearson's linear regression analyses between the average concentration of transparent exopolymeric particles [TEP ( $\mu$ g GX L-1)] and the physical, chemical, and biological parameters from each mesocosm (M1, M2, M3) and outside waters (O) divided into the two postfertilization phases of the VAHINE experiment. P1 = days 5-14, P2 = days 15-23. Each TEP value is an average of the measurements from three sampled depths. Correlations in bold are statistically significant with P < 0.05. For Het-1 and UCYN-C the growth rate ( $\mu$ ) is the net growth rate, based on changes of nifH copies L-1 from day to day.

| Parameter               | Mesocosm | Period | R2     | Р     | n |
|-------------------------|----------|--------|--------|-------|---|
|                         | M1       |        | 0.055  | 0.577 | 8 |
|                         | M2       | D1     | 0.015  | 0.776 | 8 |
|                         | M3       | PI     | 0.191  | 0.279 | 8 |
| Temperature             | 0        |        | 0.156  | 0.381 | 7 |
| (°C)                    | M1       |        | 0.369  | 0.148 | 7 |
|                         | M2       | P2     | 0.087  | 0.520 | 7 |
|                         | M3       |        | 0.357  | 0.157 | 7 |
|                         | 0        |        | 0.001  | 0.955 | 5 |
|                         | M1       |        | 0.011  | 0.805 | 8 |
|                         | M2       | D1     | 0.055  | 0.544 | 9 |
|                         | M3       | F I    | 0.295  | 0.163 | 8 |
| DIP                     | 0        |        | 0.038  | 0.677 | 7 |
| (µmol L -1 ) | M1       |        | 0.031  | 0.676 | 8 |
|                         | M2       | P2     | 0.539  | 0.038 | 8 |
|                         | M3       |        | 0.249  | 0.123 | 8 |
|                         | 0        |        | 0.171  | 0.415 | 6 |
|                         | M1       |        | 0.000  | 0.965 | 8 |
|                         | M2       | P1     | 0.198  | 0.229 | 9 |
|                         | M3       |        | 0.004  | 0.879 | 8 |
| DOP                     | 0        |        | 0.042  | 0.658 | 7 |
| (µmol L -1 ) | M1       |        | 0.128  | 0.383 | 8 |
|                         | M2       | D2     | 0.367  | 0.112 | 8 |
|                         | M3       | P2     | 0.141  | 0.320 | 9 |
|                         | 0        |        | 0.705  | 0.075 | 5 |
|                         | M1       |        | 0.020  | 0.738 | 8 |
|                         | M2       | D1     | 0.050  | 0.563 | 9 |
|                         | M3       | 11     | 0.039  | 0.641 | 8 |
| POP                     | 0        |        | 0.145  | 0.399 | 7 |
| (µmol L -1 ) | M1       |        | 0.103  | 0.401 | 9 |
|                         | M2       | P2     | 0.005  | 0.851 | 9 |
|                         | M3       | F2     | 0.192  | 0.237 | 9 |
|                         | 0        |        | 0.0005 | 0.968 | 6 |

|                                          | M1 | D1  | 0.077 | 0.51  | 8 |
|------------------------------------------|----|-----|-------|-------|---|
|                                          | M2 |     | 0.012 | 0.775 | 9 |
|                                          | M3 | PI  | 0.043 | 0.620 | 8 |
|                                          | 0  |     | 0.073 | 0.557 | 7 |
| $I_{DIP}(d)$                             | M1 | P2  | 0.238 | 0.182 | 9 |
|                                          | M2 |     | 0.523 | 0.028 | 9 |
|                                          | M3 |     | 0.338 | 0.100 | 9 |
|                                          | 0  |     | 0.239 | 0.325 | 6 |
|                                          | M1 |     | 0.155 | 0.294 | 9 |
|                                          | M2 | D1  | 0.432 | 0.077 | 8 |
|                                          | M3 | PI  | 0.048 | 0.638 | 7 |
| APA                                      | 0  |     | 0.075 | 0.553 | 7 |
| (nmole L -1 h -1 ) | M1 |     | 0.173 | 0.265 | 9 |
|                                          | M2 | D2  | 0.683 | 0.011 | 8 |
|                                          | M3 | P2  | 0.300 | 0.126 | 9 |
|                                          | 0  |     | 0.281 | 0.280 | 6 |
|                                          | M1 |     | 0.005 | 0.879 | 7 |
|                                          | M2 | D1  | 0.003 | 0.882 | 9 |
|                                          | M3 | F I | 0.051 | 0.591 | 8 |
| DOC                                      | 0  |     | 0.036 | 0.686 | 7 |
| $(\mu mol L^{-1})$                       | M1 |     | 0.266 | 0.295 | 6 |
|                                          | M2 | P2  | 0.268 | 0.482 | 4 |
|                                          | M3 |     | 0.285 | 0.275 | 6 |
|                                          | 0  |     | 0.008 | 0.888 | 5 |
|                                          | M1 |     | 0.213 | 0.211 | 9 |
|                                          | M2 | P1  | 0.005 | 0.853 | 9 |
|                                          | M3 | 11  | 0.216 | 0.246 | 8 |
| POC                                      | 0  |     | 0.099 | 0.493 | 7 |
| (µmol L -1 )                  | M1 |     | 0.006 | 0.883 | 6 |
|                                          | M2 | P2  | 0.212 | 0.358 | 6 |
|                                          | M3 | ſĹ  | 0.911 | 0.046 | 4 |
|                                          | 0  |     | 0.014 | 0.883 | 4 |
|                                          | M1 |     | 0.105 | 0.434 | 8 |
| TOC                                      | M2 | P1  | 0.003 | 0.883 | 9 |
| (µmol L -1 )                  | M3 |     | 0.002 | 0.926 | 8 |
|                                          | 0  |     | 0.006 | 0.869 | 7 |

|                                         | M1 |    | 0.745  | 0.012 | 7 |
|-----------------------------------------|----|----|--------|-------|---|
|                                         | M2 | D2 | 0.728  | 0.007 | 8 |
|                                         | M3 | P2 | 0.582  | 0.046 | 7 |
|                                         | 0  |    | 0.222  | 0.422 | 5 |
|                                         | M1 |    | 0.112  | 0.417 | 8 |
|                                         | M2 | DI | 0.042  | 0.597 | 9 |
|                                         | M3 | PI | 0.041  | 0.632 | 8 |
| DON                                     | 0  |    | 0.037  | 0.677 | 7 |
| (µmol L -1 )                 | M1 |    | 0.166  | 0.316 | 8 |
|                                         | M2 | D2 | 0.718  | 0.008 | 8 |
|                                         | M3 | P2 | 0.379  | 0.104 | 8 |
|                                         | 0  |    | 0.061  | 0.638 | 6 |
|                                         | M1 |    | 0.381  | 0.103 | 8 |
|                                         | M2 | DI | 0.160  | 0.286 | 9 |
|                                         | M3 | PI | 0.334  | 0.133 | 8 |
| PON                                     | 0  |    | 0.084  | 0.527 | 7 |
| (µmol L -1 )                 | M1 |    | 0.000  | 0.990 | 6 |
|                                         | M2 | D2 | 0.330  | 0.233 | 6 |
|                                         | M3 | P2 | 0.036  | 0.720 | 6 |
|                                         | 0  |    | 0.232  | 0.519 | 4 |
|                                         | M1 |    | 0.041  | 0.629 | 8 |
|                                         | M2 | D1 | 0.325  | 0.140 | 8 |
|                                         | M3 | PI | 0.007  | 0.858 | 7 |
| N 2 fixation                 | 0  |    | 0.0002 | 0.980 | 6 |
| (nmol L -1 d -1 ) | M1 |    | 0.046  | 0.579 | 9 |
|                                         | M2 | D2 | 0.038  | 0.617 | 9 |
|                                         | M3 | P2 | 0.405  | 0.065 | 9 |
|                                         | 0  |    | 0.267  | 0.293 | 6 |
|                                         | M1 |    | 0.251  | 0.169 | 9 |
|                                         | M2 | D1 | 0.080  | 0.460 | 9 |
|                                         | M3 | r1 | 0.054  | 0.581 | 8 |
| Chlorophyll a                           | 0  |    | 0.056  | 0.609 | 7 |
| $(\mu g \ L^{\text{-l}})$               | M1 |    | 0.096  | 0.418 | 9 |
|                                         | M2 | D2 | 0.126  | 0.348 | 9 |
|                                         | M3 | Γ∠ | 0.292  | 0.133 | 9 |
|                                         | 0  |    | 0.057  | 0.649 | 6 |

|                                              | M1 | P1 | 0.078 | 0.504 | 8   |
|----------------------------------------------|----|----|-------|-------|-----|
|                                              | M2 |    | 0.046 | 0.577 | 9   |
|                                              | M3 |    | 0.209 | 0.254 | 8   |
| PP                                           | 0  |    | 0.029 | 0.713 | 7   |
| (µmol C L -1 d -
1) | M1 |    | 0.000 | 0.991 | 8   |
| ,                                            | M2 | D2 | 0.332 | 0.105 | 9   |
|                                              | M3 | P2 | 0.124 | 0.392 | 8   |
|                                              | 0  |    | 0.499 | 0.117 | 6   |
|                                              | M1 |    | 0.083 | 0.488 | 8   |
|                                              | M2 | D1 | 0.000 | 0.973 | 9   |
|                                              | M3 | P1 | 0.549 | 0.035 | 8   |
| BB                                           | 0  |    | 0.266 | 0.236 | 7   |
| $(ngC L^{-1} h^{-1})$                        | M1 |    | 0.574 | 0.029 | 8   |
|                                              | M2 | D2 | 0.424 | 0.058 | 9   |
|                                              | M3 | P2 | 0.567 | 0.031 | 8   |
|                                              | 0  |    | 0.153 | 0.444 | 6   |

[revised manuscript text omitted]
 2015)-                                                                                                                                                                                                                                                                                                                                                                                                                                                                                                                                                                                                                                                                                                                                                                                                                                                                                                                                                                                                                                                                                                                                                                                                                           | Forma |
|                                                                                                                                                                                                               |                                                                                                                                                                                                                                                                                                                                                                                                                                                                                                                                                                                                                                                                                                                                                                                                                                                                                                                                                                                                                                                                                                                                                                                                                                                                                                             |       |
| 143                                                                                                                                                                                                           | 2.2 TEP quantification                                                                                                                                                                                                                                                                                                                                                                                                                                                                                                                                                                                                                                                                                                                                                                                                                                                                                                                                                                                                                                                                                                                                                                                                                                                                                      |       |
| 143
144                                                                                                                                                                                                    | <li>2.2 TEP quantification</li><li>Water samples (100 mL) were gently (< 150 mbar) filtered through a 0.45 μm</li>                                                                                                                                                                                                                                                                                                                                                                                                                                                                                                                                                                                                                                                                                                                                                                                                                                                                                                                                                                                                                                                                                                                                                                              |       |
| 143
144
145                                                                                                                                                                                             |  <li>2.2 TEP quantification</li> <li>Water samples (100 mL) were gently (< 150 mbar) filtered through a 0.45 μm</li> <li>polycarbonate filters (GE Water & Process Technologies). Filters were then stained with a</li>                                                                                                                                                                                                                                                                                                                                                                                                                                                                                                                                                                                                                                                                                                                                                                                                                                                                                                                                                                                                                                                                     |       |
| 143
144
145
146                                                                                                                                                                                      | 2.2 TEP quantification
Water samples (100 mL) were gently (< 150 mbar) filtered through a 0.45 μm
polycarbonate filters (GE Water & Process Technologies). Filters were then stained with a
solution of 0.02% Alcian Blue (AB) and 0.06% acetic acid (pH of 2.5). The excess dye was                                                                                                                                                                                                                                                                                                                                                                                                                                                                                                                                                                                                                                                                                                                                                                                                                                                                                                                                                                                                               |       |
| 143
144
145
146
147                                                                                                                                                                               | 2.2 TEP quantification
Water samples (100 mL) were gently (< 150 mbar) filtered through a 0.45 μm polycarbonate filters (GE Water & Process Technologies). Filters were then stained with a solution of 0.02% Alcian Blue (AB) and 0.06% acetic acid (pH of 2.5). The excess dye was removed by a quick deionized water rinse. Filters were then immersed in sulfuric acid (80%)                                                                                                                                                                                                                                                                                                                                                                                                                                                                                                                                                                                                                                                                                                                                                                                                                                                                                                                  |       |
| 143
144
145
146
147
148                                                                                                                                                                        | 2.2 TEP quantification
Water samples (100 mL) were gently (< 150 mbar) filtered through a 0.45 μm
polycarbonate filters (GE Water & Process Technologies). Filters were then stained with a
solution of 0.02% Alcian Blue (AB) and 0.06% acetic acid (pH of 2.5). The excess dye was
removed by a quick deionized water rinse. Filters were then immersed in sulfuric acid (80%)
for 2 h, and the absorbance at 787 nm was measured spectrophotometrically (CARY 100,                                                                                                                                                                                                                                                                                                                                                                                                                                                                                                                                                                                                                                                                                                                                                                                                                 |       |
| 143

149                                                                                                                                                                 | 2.2 TEP quantification Water samples (100 mL) were gently (< 150 mbar) filtered through a 0.45 µm polycarbonate filters (GE Water & Process Technologies). Filters were then stained with a solution of 0.02% Alcian Blue (AB) and 0.06% acetic acid (pH of 2.5). The excess dye was removed by a quick deionized water rinse. Filters were then immersed in sulfuric acid (80%) for 2 h, and the absorbance at 787 nm was measured spectrophotometrically (CARY 100, Varian). AB was calibrated using a purified polysaccharide GX (Passow and Alldredge,                                                                                                                                                                                                                                                                                                                                                                                                                                                                                                                                                                                                                                                                                                                                                  |       |
| <ol> <li>143</li> <li>144</li> <li>145</li> <li>146</li> <li>147</li> <li>148</li> <li>149</li> <li>150</li> </ol>                                                                                            | 2.2 TEP quantification
Water samples (100 mL) were gently (< 150 mbar) filtered through a 0.45 μm polycarbonate filters (GE Water & Process Technologies). Filters were then stained with a solution of 0.02% Alcian Blue (AB) and 0.06% acetic acid (pH of 2.5). The excess dye was removed by a quick deionized water rinse. Filters were then immersed in sulfuric acid (80%) for 2 h, and the absorbance at 787 nm was measured spectrophotometrically (CARY 100, Varian). AB was calibrated using a purified polysaccharide GX (Passow and Alldredge, 1995). TEP concentrations (μg gum xanthan [GX] equivalents L -1 ) were measured according                                                                                                                                                                                                                                                                                                                                                                                                                                                                                                                                                                                                                                   |       |
|  <li>143</li> <li>144</li> <li>145</li> <li>146</li> <li>147</li> <li>148</li> <li>149</li> <li>150</li> <li>151</li>                                                                                | 2.2 TE